



# Soil hydraulic and hydrological data from seven field sites in the Thames catchment, UK, 2021

John Robotham[1,2,a], Emily Trill[1,b], James Blake[1], Ponnambalam Rameshwaran[1], Peter Scarlett[1], Gareth Old[1], Joanna Clark[2,c]

[1]UK Centre for Ecology & Hydrology, Wallingford, OX10 8BB, UK
[2]Department of Geography and Environmental Science, University of Reading, Reading, RG6 6AB, UK
[a]now at: Environment Agency, Wallingford, OX10 8BD, UK
[b]now at: Thomson Environmental Consultants, Cardiff, CF11 9LJ, UK
[c]deceased

*Correspondence to*: James Blake (jarib@ceh.ac.uk)

**Abstract.** Observational data of soil physical and hydraulic properties are important for improving our understanding of hydrological processes. This is particularly relevant given current interest in the potential of land-based "natural flood management" measures (and related concepts: "nature-based solutions" and "working with natural processes") to reduce flood risk. Therefore, a detailed survey of seven field sites under different land-uses and management practices in the Thames catchment, UK, was undertaken as part of the "LANDWISE" project. Measurements (n = 1300) included soil bulk density, estimated porosity, soil moisture and soil moisture retention, surface infiltration rate, and saturated hydraulic conductivity. Field sites comprised three arable fields on shallow soils over Limestone, two arable fields on free draining loamy soils over Chalk, and permanent grassland and broadleaf woodland on slowly permeable soil over Mudstone. Soil sampling points covered infield areas, trafficked areas (e.g. tramlines), and untrafficked margins. Samples were generally taken at five depths ranging from the soil surface to 100 cm below ground level. Soil saturated hydraulic conductivity measurements were made at 25 and 45 cm depths. Soil samples and measurements were taken between April and October 2021, with repeats taken pre- and post-harvest (arable sites). These data provide valuable insight into the hydrological behaviour of soils under contrasting management, including both conventional and innovative agricultural practices (e.g. herbal leys, mob grazing and controlled traffic). Dataset applications include: improving the performance of hydrological and land surface models, and validation of remotely-sensed soil observations. The dataset is publicly available at https://doi.org/10.5285/a32f775b-34dd-4f31-aafa-f88450eb7a90 (Trill et al., 2022).

## 1 Introduction

The physical and hydraulic properties of soil play crucial roles in controlling the extent to which soils are able to absorb, store, and transmit water (Cleophas et al., 2022; Vogelmann et al., 2013; Nimmo, 2005). Such properties include soil texture, porosity, pore size distribution and connectivity, soil moisture retention and hydraulic conductivity. These properties, together



with antecedent soil moisture conditions, exert a critical control on the partitioning of rainfall into infiltration and runoff, with implications for flood risk. They are also important in determining the fate of pollutants, and the accessibility of water for plants and crops, thus affecting a range of ecosystem services and biogeochemical processes (Indoria et al., 2020; Fu et al.,

2021). Previous studies have shown how soil hydraulic properties can exhibit high spatial variability at regional, catchment, and even field scales (Usowicz and Lipiec, 2021; Peck et al., 1977). This makes observational data on such properties valuable for improving our understanding of hydrological processes, and also for being able to accurately represent these processes within models. Soil hydraulic properties have been widely used in pedotransfer functions which relate these properties to more easily determined properties such as soil texture (Lilly, 2000; Cooper et al., 2021; Schaap et al., 2001; Wösten et al., 1999;

Hollis et al., 2015). This has helped to overcome some of the issues surrounding the scarcity of data on soil hydraulic properties, and by doing so provide better model predictions of processes such as water flow and solute transport at larger scales (Vereecken et al., 2010). However, this does not negate the need for new observational data, particularly with respect to the influence of land-use and management on soils and their hydrological functioning.

Over the past decades, growing concern over the need for sustainable agriculture and delivery of ecosystem services has led

to greater interest in the effect of innovative land management on soil properties (Mihelič et al., 2020; Blanco-Canqui and Ruis, 2018; Haruna et al., 2018; Martínez-Mena et al., 2020; Terefe et al., 2020; Hartmann and Six, 2022; Page et al., 2020; Soane et al., 2012; Antille et al., 2019; Strudley et al., 2008). Land-use and land management have been shown to influence below-ground water storage and movement (Niu et al., 2015), as well as surface run-off generation processes (Germer et al., 2010). In terms of agriculture, the choice of cropping system (e.g. mixed cropping) has been shown to influence soil properties

such as porosity (Haynes and Francis, 1990). There is evidence to suggest that modifying tillage practices can improve the soil hydro-physical properties (e.g. bulk density and hydraulic conductivity) of arable fields. For example, Singh et al. (2022) found that zero tillage showed the best potential to effectively reduce run-off generation when compared with conventional tillage and minimum tillage. Another area of concern for maintaining healthy soil on agricultural land is compaction and its impact on soil structure, particularly as a result of traffic from heavy machinery such as tractors (Wang et al., 2022). Innovative

approaches such as controlled traffic farming aim to minimise potential soil compaction by confining it to specific areas and thereby reducing the overall area of field subjected to farm traffic (Shaheb et al., 2021). There is evidence to suggest that applying such practices can help to restore soil structure, increase water holding capacity and macropore density, and reduce bulk density (McHugh et al., 2009).

The "LANDWISE" ("LAND management in loWland catchments for Integrated flood riSk rEduction") project aimed to

evaluate the effectiveness of realistic and scalable land-based Natural Flood Management (NFM) measures to reduce the risk from surface water, river, and groundwater flooding in lowland catchments. This research focussed on NFM solutions related to how land and soil is managed (e.g. crop choice and tillage practices) in order to fill gaps in the current knowledge surrounding their effect on soil hydrology and potential effectiveness for flood mitigation at different scales (Dadson et al., 2017). Measurements were made across different land uses, land management practices and soil types within the Thames

catchment, UK, to investigate infiltration and below-ground water storage.

An initial LANDWISE broad-scale field survey measured a large sample (n = 1836) of near-surface physical and hydrological soil properties across 164 fields/land parcels in the Thames catchment which represented four broad land-use and management classes: arable with and without grass in the crop rotation, permanent grassland, and broadleaf woodland. These field locations also covered five generalised soil/geology classes, and covered a range of conventional and innovative agricultural practices.

Data and the details for the broad-scale survey are available separately to this paper; see Blake et al. (2022). Following on from the broad-scale campaign, a LANDWISE detailed field survey was undertaken to better understand the variability of soil properties with depth and over time. This paper describes the data collected during this detailed survey which focussed on a subset of seven fields from the broad-scale study (Fig. 1). Key variables within the dataset include: soil dry bulk density (n = 543), estimated porosity (n = 543), soil moisture (n = 537) and soil moisture retention (n = 38), surface infiltration rate (n =

117), and saturated hydraulic conductivity (n = 65).

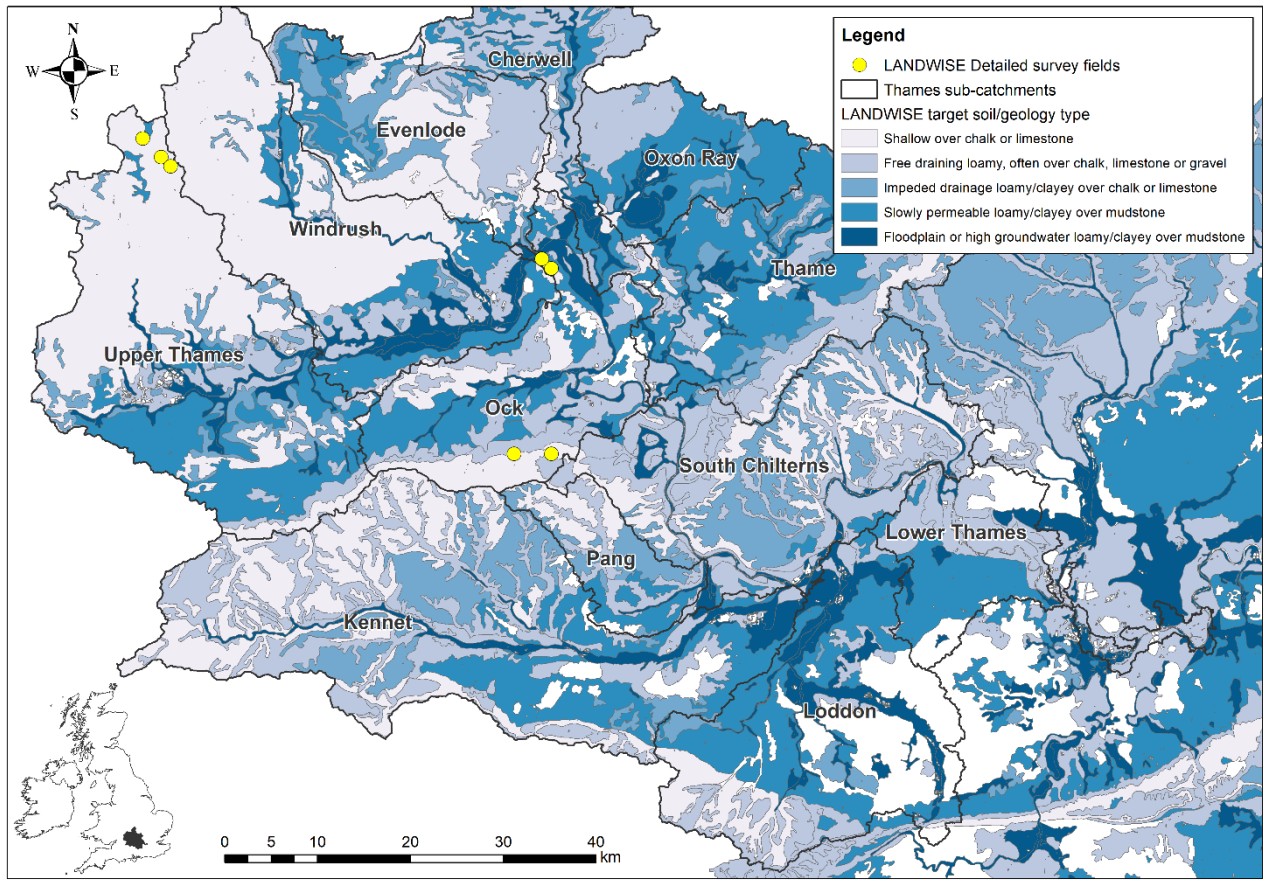

**Figure 1: Locations and soil/geology types of the LANDWISE Detailed survey field sites within the Thames catchment, UK. Field locations are displayed to the nearest 1 km for participant anonymity purposes. See Table 1 for further details. Soils Data © Cranfield University (NSRI) and for the Controller of HMSO 2023. Map colours based on ColourBrewer: Color Advice for Maps (2023) by**
**Cynthia A. Brewer, Geography, Pennsylvania State University.**



## 2 Methodology

### 2.1 Field Sites

The seven LANDWISE detailed survey field sites were selected based on their soil type and underlying geology, land-use,
and management (as detailed in Table 1). Two field sites were located (on similar soil types, based on data from Blake et al., 2022) over Mudstone, two over Chalk, and three over Limestone. Each of these site groupings allows for comparisons between land-uses or types of management. The Mudstone sites compare grazed permanent grassland and mature broadleaf woodland. The Chalk sites compare arable rotations with and without controlled machinery traffic. The Limestone sites compare conventional ploughed arable farming to no till arable farming with grass or herbal leys in the rotation. For ease of reference
herein, each survey site has been given a brief 'Site/Field Name' based on the underlying geology type, e.g. "Chalk 1".



**Table 1. LANDWISE detailed survey field sites, showing location, land-use, management, soil and geology.**

| Site and field ID[a] | Location[b] | Land-use[c] | Crop (rotation) or vegetation[d] | Land management (tillage, cover crops, traffic etc.)[d] | Soil type (LANDWISE survey grouping[f]) | Soil survey data (hand texture and reaction to HCl)[g] | Bedrock geology[h] | Site/Field Name[i] |
|---|---|---|---|---|---|---|---|---|
| 18_6 | 447000, 208000 | Grassland, permanent | Perennial ryegrass (mainly) | Cattle or sheep, mob grazing | Slowly permeable loamy / clayey over Mudstone | Texture: SaSiLo HCl: none-slight | Mudstone | Mud 1 |
| 44_1 | 446000, 209000 | Broadleaf woodland, mature | Mixed native broadleaf species (ash, sycamore, hawthorn) | Unmanaged, low density grazing deer | | Texture: SiLo HCl: none | | Mud 2 |
| 23_5 | 447000, 189000 | Arable without grass | 2015 Oilseed rape, 2016 Winter wheat, 2017 Spring barley, 2018 Spring beans, 2019 Winter wheat, 2020 Winter barley, 2021 Oilseed rape | Conventional[e], controlled traffic, min till, cover crops | Shallow over Chalk | Texture: SiCl HCl: strong | Chalk | Chalk 1 |
| 26_1 | 443000, 188000 | Arable with grass | 2019 Spring barley, 2020/21 Grass, 2022 Winter wheat, 2023 Oilseed rape | Conventional[e], min till before grass, grass to be ploughed | | Texture: SiCl – SiClLo HCl: strong | | Chalk 2 |
| 31_3 | 403000, 222000 | Arable without grass | 2019 Barley, 2020 Oats, 2021 Winter wheat | Conventional[e], conventional till (ploughed) | Shallow over Limestone | Texture: SiLo HCl: strong | Limestone | Lime 1 |
| 27_2 | 405000, 220000 | Arable with grass (ryegrass and clover) | 2016/17/18/19 Grass and Clover Ley, 2020 and 2021 Winter wheat | Conventional[e], no till | | Texture: SiLo – SaSiLo HCl: strong | | Lime 2 |
| 27_4 | 406000, 219000 | Arable with grass (herbal ley) | 2016/17/18 Herbal Ley, 2019 Winter wheat, 2020 Peas, 2021 Winter wheat | Conventional[e], no till | | Texture: SiLo HCl: slight-strong | | Lime 3 |

[a] ID code corresponding to "ID_SiteNo_FieldNo" in dataset

[b] British National Grid Reference (Easting, Northing). Due to participant anonymity requirements, locations are given to the nearest 1 km.

[c] LANDWISE target land use

[d] Data from land owner/manager survey and/or field observations. Crop during period of sampling underlined

[e] "Conventional" indicates non-organic farming

[f] LANDWISE survey soil types derived from Cranfield "Soilscapes" classes: Soils Data © Cranfield University (NSRI) and for the Controller of HMSO 2023



[g] Additional soil data are dominant values from LANDWISE Broadscale field survey (Blake et al., 2022). Soil texture class abbreviations: Sa(nd), Si(lt), Lo(am), Cl(ay).

[h] Indicative bedrock geology from DiGMapGB data at 1:625 000 scale: Reproduced with the permission of the British

Geological Survey ©NERC. All rights Reserved

[i] Brief "Site/Field Name" for use in this paper



## 2.2 Sampling Strategy

The sampling strategy of the LANDWISE detailed survey was designed so that fields were sampled across multiple periods over an annual cycle. Table 2 lists the measurements made and the techniques used, along with the number of samples taken and when they were taken in the farming calendar.

Soil properties were measured along transects within fields at different location types that were categorised into: TR (trafficked areas e.g. tramlines), IN (general infield areas), and UN (untrafficked field margins). Each of the three location types was

sampled five times within each field, giving a total of 15 locations per field (see Fig. 2 for details). The UN sample transect was necessarily located along the field margin, with the two other transects offset into the field with 30 m spacing from both each other and from the UN transect. Transect locations and spacing were selected to avoid potentially excessively compacted soil in the vicinity of field gateways and to avoid sampling within 30 m of the field margins to avoid any current or historic turning headlands. Within field transects were located approximately perpendicular to any tramlines. Sampling locations were

spaced approximately evenly along each transect, although due to practical constraints not all soil measurements were made at each of these locations (see Table 3). Once in the general vicinity of the sampling location, field survey discretion was used to ensure that a representative sampling location was selected. The five IN locations were chosen to represent general infield conditions typical of the majority of the field area. The five TR locations were targeted to the trafficked parts of the field, typically tramlines in arable fields, and animal/livestock tracks in grassland/woodland. The five UN locations were chosen on

field margins that were uncultivated and untrafficked, avoiding any margin trafficking associated with hedge cutting or other activities and also avoiding sampling within 1 m of tree/hedge stems and animal burrows. In the woodland, sampling locations were positioned along two transects at least 1 m away from the base of any trees.





**Table 2. Summary of soil measurements, techniques used for sampling, and timing of sampling for the LANDWISE detailed survey.**

| Measurement | Technique | Site type | Samples per field | No. of fields | Spring | Pre-harvest | Post-harvest |
|---|---|---|---|---|---|---|---|
| Soil bulk density (and inferred porosity) and volumetric water content with depth | Volumetric soil cores and oven drying | Arable | 50 | 5 | April 2021 | NA | October 2021 |
| | | Grassland | 50 | 1 | | | |
| | | Woodland | 50 | 1 | | | |
| Soil moisture retention (pF 0 – pF 2) | Volumetric samples and Sandbox lab analysis | Arable | 24 | 3 (Limestone sites only) | October 2021 | | |
| Soil field saturated hydraulic conductivity | Guelph Permeameter | Arable | 12 | 5 | September-October 2021 | | |
| | | Grassland | 12 | 1 | | | |
| | | Woodland | 12 | 1 | | | |
| Soil infiltration rate (at 0.5 or 2.0 cm suction) | Mini Disk Infiltrometer | Arable | 10 | 5 | NA | July 2021 | September 2021 |
| | | Grassland | 10 | 1 | | | |
| | | Woodland | 10 | 1 | | | |
| Soil and vegetation root depth | Auger and tape measure | Arable | 15 | 5 | April 2021 | NA | October 2021 |
| | | Grassland | 15 | 1 | | | |
| | | Woodland | 15 | 1 | | | |




**Table 3. Sampling locations used for LANDWISE detailed survey soil property measurements.**

| Soil Measurement | Sampling locations | | |
|---|---|---|---|
| Bulk density (and inferred porosity) and volumetric water content | TR1 - TR5 | IN1 - IN5 | - |
| Moisture retention | TR2 - TR4 | IN2 - IN4 | - |
| Saturated hydraulic conductivity | TR2 - TR4 | IN2 - IN4 | - |
| Infiltration rate | - | IN1 - IN5 | UN1 - UN5 |
| Soil and vegetation root depth | TR1 - TR5 | IN1 - IN5 | UN1 - UN5 |


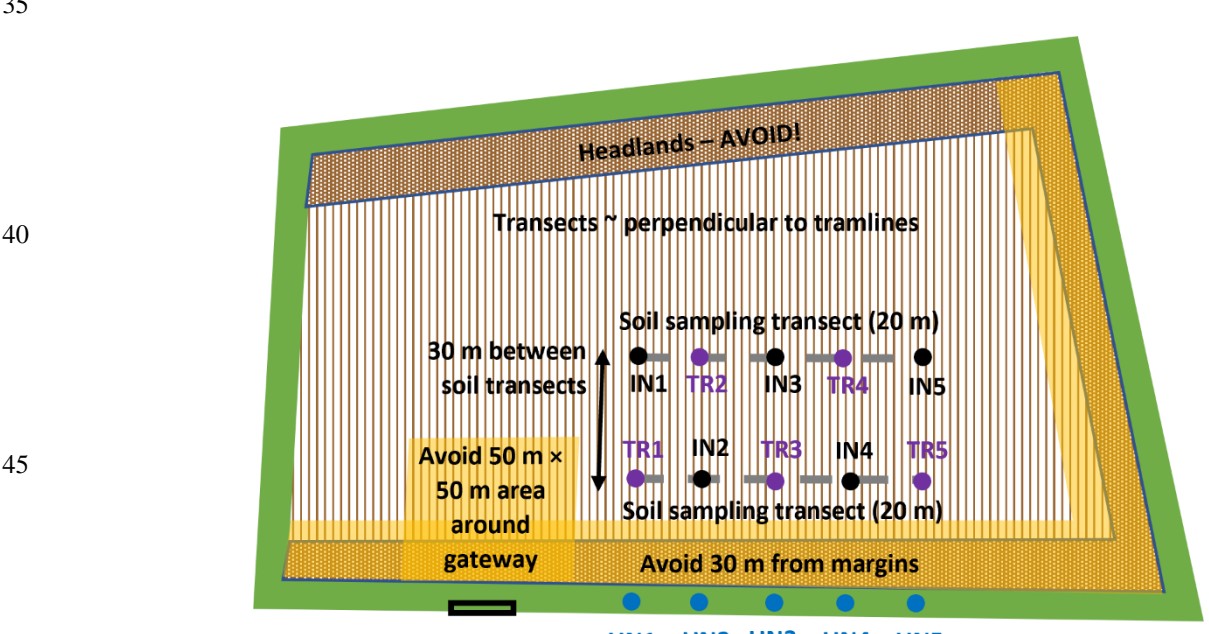



Figure 2: Schematic of the LANDWISE detailed survey transects and sampling locations within a typical field. IN = Infield; TR = Trafficked; UN = Untrafficked margin.

## 2.3 Soil and Vegetation Root Depth

Soil depth profiles and the root depths of vegetation were examined at each sampling location. First, an Eijkelkamp gouge

auger was used to sample the top 50 cm of soil (where possible, depending on the soil type and geology). A tape measure was

then used to measure the depths of distinct soil horizon boundaries (e.g. at the bottom of the plough layer) and the maximum

rooting depth where this was shallower than 50 cm. The gouge auger was then re-inserted to sample the next 50 cm of soil,





and measure any soil horizon boundaries and the rooting depth. Where the soil depth extended beyond 1 m, this augering and measuring process was continued until no longer possible.

## 2.4 Volumetric Water Content, Bulk Density and Porosity


Soil samples were collected using an Eijkelkamp 07.53.SC sample ring kit with closed ring holder (50 mm diameter rings; 100 cm³ volume) combined with an Edelman auger or Stony auger as required (Royal Eijkelkamp; Giesbeek, The Netherlands). Following careful removal of any loose surface vegetation, soil samples were taken at depths of 0-5 cm, 10-15 cm, 25-30 cm, 45-50 cm, and 95-100 cm below ground level (bgl). Due to limited soil depths, some of the deeper samples were not always

possible. Upon extraction, samples were carefully trimmed using a soil spatula to ensure volume control, placed in a polythene bag and sealed. Samples were returned to the laboratory on the day of collection and kept refrigerated at 4 °C prior to analysis. Volumetric water content (VWC) was obtained by oven drying the volumetric soil samples. Whilst samples were still in their bags, any particularly large soil aggregates were manually broken up to aid drying. Samples were emptied into pre-weighed aluminium foil trays and then weighed to the nearest 0.1 g. The sample trays were then placed into a drying oven at 105 °C

for approximately 36 hours (up to a maximum of 60 hours). Upon completion of drying, samples were removed from the oven in small batches and immediately weighed (to the nearest 0.1 g) to ensure they did not regain any moisture from the atmosphere. VWC (cm³ cm⁻³) was then calculated using Eq. (1-2):

$$VWC = \frac{v_w}{v} \qquad (1)$$

Where $v_w$ is the volume of water (cm³), and $v$ is the sample volume (cm³; equal to 100.1 cm³).

$$v_w = \frac{m_{wet} - m_{dry}}{\rho_w} \qquad (2)$$

where $v_w$ is the volume of water (cm³), $m_{wet}$ is the wet soil mass (g), $m_{dry}$ is the dry soil mass (g), and $\rho_w$ is the density of water (g cm⁻³; equal to 1 g cm⁻³).

Dry bulk density (BD) was then calculated using Eq. (3):

$$\rho_{\text{bulk\_dry}} = \frac{m_{dry}}{v} \qquad (3)$$

where $\rho_{\text{bulk\_dry}}$ is soil BD (g cm⁻³), $m_{dry}$ is the dry soil mass (g), and $v$ is the sample volume (cm³).

Soil porosity was estimated using Eq. (4):

$$\varphi_{est} = 1 - \frac{\rho_{\text{bulk\_dry}}}{\rho_{\text{mineral}}}, \qquad (4)$$

where $\varphi_{est}$ is the estimated soil porosity (cm³ cm⁻³), $\rho_{\text{bulk\_dry}}$ is the soil dry bulk density (g cm⁻³), and $\rho_{\text{mineral}}$ is the particle density of mineral soil (g cm⁻³) which was assumed to be 2.65 g cm⁻³, as commonly used in soil science, e.g. Blake (2008).



### 2.5 Soil Moisture Retention

Soil samples for soil moisture retention analysis were collected following a procedure similar to that given in Sect. 2.4, except that the collected samples were retained in their soil sampling ring and capped prior to analysis. These samples were taken from 0-5 cm, 25-30 cm, and 45-50 cm bgl, where the soil profile allowed. Care was taken to ensure that the sample soil surfaces were not smeared by the sample trimming process. The laboratory analysis was carried out using an Eijkelkamp Sandbox 08.01 (Royal Eijkelkamp; Giesbeek, The Netherlands) that applied a range of pressures (pF 0 to pF 2; from saturation to 100 cm suction) to soil samples.

Small pieces of thin permeable nylon cloth were attached to the base of the soil samples (still contained within metal sample rings) using rubber rings, in order to retain the soil samples as they were placed on and removed from the sandbox surface whilst being weighed during the analysis process. Each cloth and rubber ring were weighed together before use and the sample was then reweighed once the cloth and ring were attached.

The samples were then spaced out evenly on the sandbox surface, ensuring a gap of several centimetres between them. Deionised water was used to supply water to the sandbox during the saturation process (with the water level raised slowly to avoid air entrapment). The water supply contained a small addition of copper sulphate to help prevent algal growth within the sandbox and its tubing. Samples were left to saturate in the sandbox until they reached a constant mass. Saturation was assumed to have been reached when sample mass (measured using a 0.1 g precision balance) remained the same for two consecutive days. Evaporation from the sandbox was minimised by placing a tight lid over the sandbox when the samples were not being weighed.

Once all samples reached saturation, the suction regulator was changed to apply a suction of 2.5 cm head for the first pF (0.4). Samples were weighed on a daily basis until they reached equilibrium (consistent mass over two consecutive days). This process was then repeated at each of the following pF values: 1, 1.5, 1.7, 1.8 and 2 (equivalent to 100 cm suction). Due to practical constraints soil moisture retention beyond pF 2.0 was not measured.

### 2.6 Infiltration Rate

Soil surface infiltration measurements were taken using Mini Disk Infiltrometers (Decagon Devices, Inc.; Pullman, Washington, USA). The infiltrometers are made up of an upper and lower chamber (both of which are filled with water during measurements). The upper chamber controls the suction, and the lower chamber contains a volume of water that infiltrates into the soil at a rate determined by the suction. The bottom of the infiltrometer contains a porous sintered stainless-steel disk that does not allow water to leak in open air, so only allows water out when placed on a relatively level soil surface. Once the infiltrometer is placed onto the soil surface for a measurement, water leaves the lower chamber and infiltrates into the soil (the rate of which is determined by the soil properties). As the water level in the lower chamber drops over time, the rate of infiltration can be calculated by recording the water volume at regular intervals (e.g. 30 seconds for a typical silt loam).

Where the soil surface was vegetated, the soil surface first had to be exposed at the sampling locations. Vegetation was carefully cleared using a hand trowel to ensure the soil remained relatively undisturbed. Reliable measurements required the soil surface to be as level as possible, so locations were chosen with this in mind. Where it was not possible to get a naturally level surface, the soil was gently brushed (avoiding any soil smearing) to create a flat surface for the infiltrometer to rest on
and ensure a good contact with the porous disk.

At each site, the suction rate of the infiltrometer was selected based on prior knowledge of the soil type (Table 1) following guidance in the Mini Disk Infiltrometer manual (Decagon, 2022). At most sites the recommended suction rate of 2 cm was used, however on the two Mudstone sites a suction rate of 0.5 cm was used due to the lower infiltration rates anticipated there. The measurement interval was also chosen based on the soil type at each site and adjusted accordingly to suit the observed
rate of infiltration.

The infiltrometer was kept upright and stable during the measurement using a retort stand and clamp. Following the recommendation in the infiltrometer manual, the chambers were filled using tap water rather than distilled water to avoid potential changes to the ionic balance of soil water and its effects on soil properties. Before taking a measurement, the initial water volume of the lower chamber was recorded and then the infiltrometer was placed onto the soil surface whilst a timer was
started. Measurements were then taken at the selected time interval. The method aimed to infiltrate at least 15 ml of water for a robust measurement, however given fieldwork time constraints this was not always practical where infiltration rates were notably slow. These measurements were still used in the dataset but were flagged during the data quality control process (see Sect. 3.1).

The recorded field data were entered into a Microsoft Excel spreadsheet (available at www.decagon.com/macro) (METER
Environment), which was used to calculate the unsaturated hydraulic conductivity ($K_{unsat}$) at the specified suction (infiltration rate). The calculations used within the spreadsheet follow the method proposed by (Zhang, 1997), which measures cumulative infiltration over time and fits the results using a curve function. The soil van Genuchten parameters (for different soil texture classes) required by the function were obtained from Carsel and Parrish (1988).

## 2.7 Saturated Hydraulic Conductivity

Soil field-saturated hydraulic conductivity ($K_{fs}$) was measured at two depths (25 and 45 cm bgl) using model 2800 Guelph Permeameters (Soilmoisture Equipment Corp.; Goleta, California, USA). The Guelph Permeameter operates using the Mariotte Principle and measures the steady-state rate of water recharge into unsaturated soil from a well hole, in which a constant head of water is maintained.

Measurements of $K_{fs}$ were taken nearby but offset from locations on field transects (Fig. 2) to avoid errors due to soil
disturbance from previous measurements and sampling. Well holes for the 25 and 45 cm depth measurements were made approximately 1 metre apart to avoid interference from soil moisture saturation 'bulbs' created during measurements. An Edelman soil auger was initially used to excavate a well to depths of 10 and 30 cm bgl, for the 25 and 45 cm measurements respectively. To remove the remaining depth of soil from the well, a sizing auger attachment was then used to ensure a well

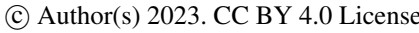

hole of a uniform geometry (6 cm in diameter with a flat bottom). Where large stones were encountered during the augering
process, new well holes were dug to avoid excessively increasing the volume of the well by removing stones from the walls
of the well hole. Once the well hole was of suitable depth and geometry, a well prep brush attachment was used to lightly
roughen the walls of the well hole and to remove any soil smearing that may have occurred during the augering process.

The Guelph Permeameter was assembled, reservoir filled with deionised water, and placed on a tripod above the well hole
following the operating instructions (Soilmoisture Equipment Corp., 2012). For the measurements, the 'two head' method was
chosen due to its higher accuracy compared to the 'single head' method. The two head method involved repeating
measurements at two well head heights. At most field sites, measurements were carried out using well head heights of 5 and
10 cm above the base of the well. However, at the two sites with more slowly permeable soils, increased heights of either 10
and 15 cm or 15 and 20 cm were used for some measurements to ensure the measurement duration remained practical given
time constraints. Similarly, where measurements were anticipated to be slow, the inner reservoir of the permeameter
(containing a smaller volume of water) was used instead of the larger combined reservoir which was used for most
measurements.

To take measurements with the permeameter, the steady-state rate of fall ($R$) was determined by recording the reservoir water
volume at regular time intervals. The rate of fall in the volume was calculated for each time interval and the measurement was
only finished when there was no significant change for three consecutive time intervals.
A Microsoft Excel spreadsheet (available at https://www.soilmoisture.com/Calculators) was used to calculate $K_{fs}$ (Guelph
Permeameter K-sat Calculator). The calculation takes into account the observed $R$ value from a measurement, the cross-
sectional area of the reservoir used, the well head height, and the well hole radius.

**3 Data**

**3.1 Data Quality Control**

Data underwent a quality control (QC) process to flag any data entry errors or potentially spurious measurements. Hardcopy
measurements made in the field and lab were entered into Excel spreadsheets by the person who recorded them. To avoid data
entry errors, the transcribed data were then checked by another individual and corrected where necessary.

Infiltration measurements underwent QC to categorise each measurement using the following data flags which are stored as
"Infiltration_1_QC_Flag" and "Infiltration_2_QC_Flag" in the dataset:

• "Good" = Where no apparent issues with the measurement were identified.

• "Invalid" = Where the measurement gave values that were not physically plausible (e.g. negative values). These
values have been removed from the dataset.

• "A" = Where the change in infiltration rate over time was observed to be notably unsteady (e.g. where plots of
cumulative infiltration over time showed sudden/rapid changes).





- "B" = Where < 15 ml water infiltrated during the measurement (the Mini Disk infiltrometer manual states that accurate calculation requires at least 15 ml of water to be infiltrated during each measurement).
- "C" = Where calculated $K_{unsat}$ values were unusually high. This was determined by comparing the measured value against typical values + 3 SD (i.e. the 99.7% upper bound of the distribution) from Carsel and Parrish (1988). It is important to note that datapoints with this QC flag may in fact be correct and potentially reflect the novel soil
state/structure/management at the time of measurement.

Saturated hydraulic conductivity measurements underwent QC to categorise each measurement using the following flags which are stored in the "Guelph_Permeameter_QC_Flag" column of the dataset:

- "Good" = Where no apparent issues with the measurement were identified.
- "Invalid" = Where measurements gave values that were not physically plausible, e.g. negative values or "alpha"
values outside of the valid range of 0.01-0.5 cm$^{-1}$ (Soilmoisture Equipment Corp., 2012). These values have been removed from the dataset.

In addition, the "Guelph_Permeameter_notes" column indicates whether the double head method or the mean of two single head measurements was used for deriving $K_{fs}$ for each measurement. The double head method is more accurate and is therefore provided in preference. However, sometimes the data generated physically inadmissible values when using the double head
method and in this case the data was instead used in two separate single head measurement calculations. The results of the two single head measurements were then averaged. Further details are given in the Guelph Permeameter operating instructions (Soilmoisture Equipment Corp., 2012).

### 3.2 Volumetric Water Content, Bulk Density and Porosity

VWC, BD and estimated porosity were determined from soil sampling at five different depths from the surface down to 100 cm
where possible. Example BD data, showing variation with depth, are illustrated in Fig. 3. All depths were sampled at Mud 1 and 2, and Chalk 2, whereas samples could only be obtained from a maximum depth of 45-50 cm at Chalk 1, Lime 2 and 3, and at 25-30 cm at Lime 1 (due to the shallow soil profiles at these sites). Overall, BD showed an increasing trend with depth, but showed signs of reaching a maximum value between around 45-50 cm and 95-100 cm bgl.

Soil sampling occurred in April and October from both infield and trafficked locations at each field. For the arable sites, the
October samples were taken post-harvest and generally pre-cultivation (if applicable). Comparing the sites over time shows that BD of the soil surface (0-5 cm) was typically higher in October than in April (Fig. 4). The highest soil surface BD were observed at the two arable sites on loamy soil underlain by Chalk, whereas the lowest occurred on the grassland and woodland sites with more clayey soils underlain by Mudstone. In terms of the sampling location types, the differences between infield and trafficked soil BD at the surface were more notable in April, with higher median BD at the arable sites. In April, the infield
areas of the Chalk 1 controlled traffic site have a substantially lower bulk density compared to the trafficked areas or indeed both area types at the nearby Chalk 2 conventional traffic site.

It is important to note that the estimated porosity data are derived purely based on the assumed density of the soil mineral particles, without accounting for the proportion or density of any soil organic matter present. Therefore, estimated porosities for the soil surface samples are likely to be less accurate than the samples at greater depth due to the greater influence of

organic matter in the topsoil. However, the LANDWISE Broadscale dataset provides adjusted estimates of soil surface porosities which also take organic matter content into account (Blake et al., 2022).

**Figure 3: Changes in mean soil dry bulk density (g cm⁻³) with depth (cm) at infield and trafficked locations sampled in April and**
**October. Error bars represent standard deviations. Plotting on "Depth" axis is at lower limit of sampled depth, e.g. 45-50 cm samples are plotted at 50 cm.**

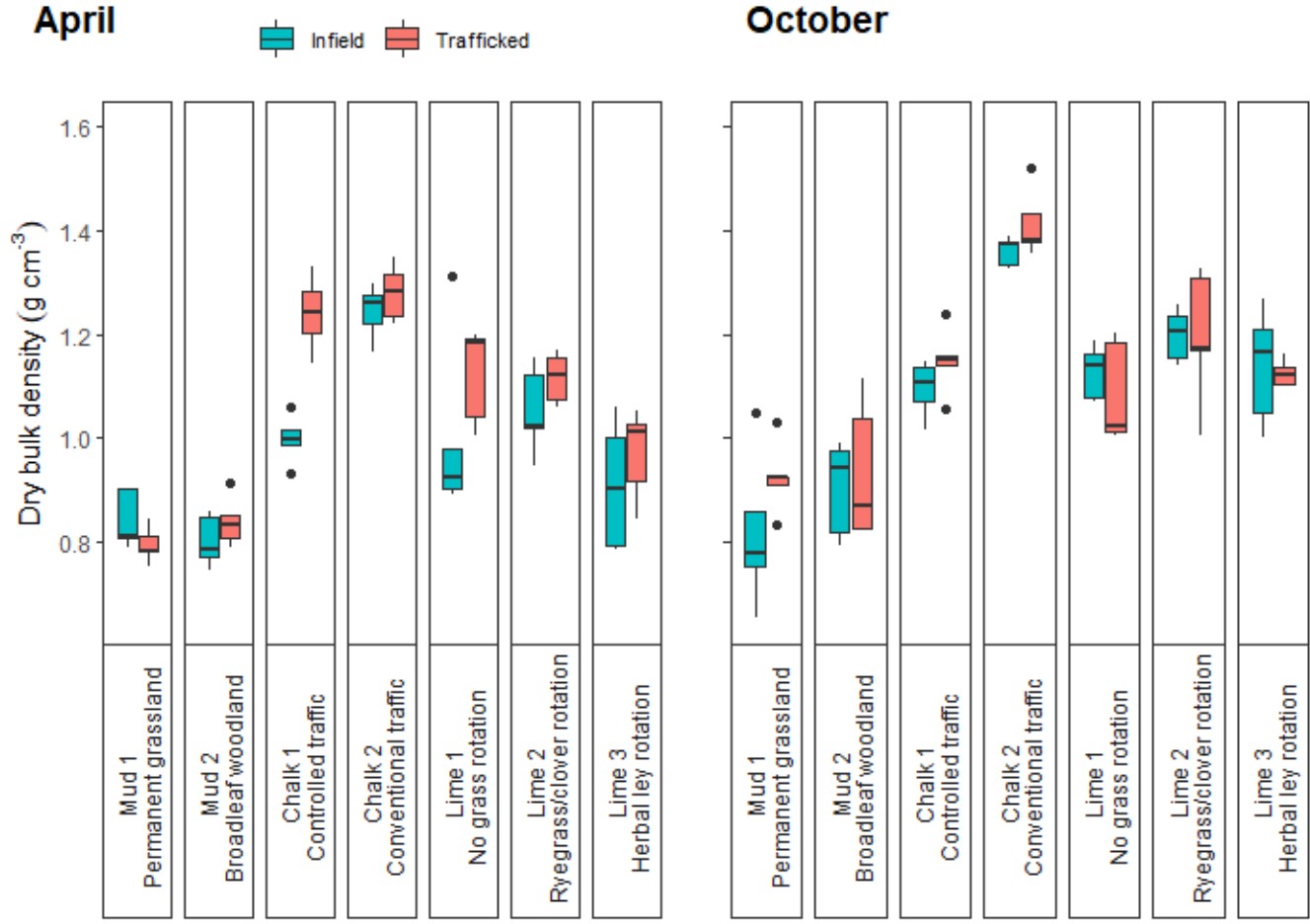

**Figure 4:** Box plots of soil surface (0-5 cm depth) dry bulk density (g cm$^{-3}$) sampled at infield and trafficked locations in April and
October (pre- and post-harvest respectively for the arable sites).

### 3.3 Soil Moisture Retention

Soil moisture retention data describe the mean average relationships between soil suction and VWC for each sampling location
type and depth. The three fields where soil moisture retention data are available are all on shallow soils over limestone, thereby
allowing comparisons to be made between the different agricultural management practices and their potential influence on soil
hydraulic properties. Soil moisture retention data show that the steepest decreases in VWC with increasing soil suction
occurred in the field with no grass in rotation, whereas the fields with ryegrass and clover in rotation, and the herbal ley rotation
had comparably more gentle decreases in VWC with increasing soil suction (Appendix A1). Considerable variation was
observed in moisture retention curves between both soil depths and location types for the herbal ley field, with VWC at
saturation (pF 0) ranging from 0.53 to 0.63 cm$^3$ cm$^{-3}$.



### 3.4 Infiltration Rate

Infiltration rate measurements are summarised in Fig. 5 as boxplots for each field, grouped according to when the measurements took place and also by the location type (infield or untrafficked margins). At the non-arable sites (Mud 1 and 2), infiltration measurements were only taken infield due to the absence of untrafficked margins. Infiltration rate varied by
several orders of magnitude, ranging from ~$1\times10^{-6}$ cm s$^{-1}$ to $5\times10^{-3}$ cm s$^{-1}$. The permanent grassland soil over Mudstone was found to have the highest median infiltration rate both in July and September. However, it should be noted that infiltration rate at the Mudstone sites cannot be directly compared to other sites due to the different suction values used for the measurements. Infiltration rates at the Mudstone sites are likely to be higher due to the lower suction allowing larger pore sizes to be activated for infiltration. For the arable soils, infiltration was on average faster in September with the exception of the untrafficked
margin at site Chalk 1, which exhibited notably low rates in comparison to the other sites.



**Figure 5: Box plots of infiltration rate (cm s⁻¹) measurements taken at infield and untrafficked margin locations in July and September (pre- and post-harvest respectively for the arable sites). Bold lines indicate median values. Note the log scale.**

## 3.5 Saturated Hydraulic Conductivity

Figure 6 summarises the soil saturated hydraulic conductivity data grouped by the location type and measurement depth. $K_{fs}$ varied by several orders of magnitude, with measurements across all field sites, location types and depths ranging from $< 1 \times 10^{-6}$ cm s⁻¹ to $> 0.03$ cm s⁻¹. Irrespective of location type and depth, the greatest variability in soil $K_{fs}$ was observed in the broadleaf woodland site (Mud 2). This variability reflects the heterogenous complexity of woodland soils in terms of both space and depth. For example, soil water movement can be heavily influenced at a local scale by features such as decayed root channels. Furthermore, the woodland site was underlain by Mudstone, with the soil texture becoming increasingly clayey with depth, and therefore likely to be a cause of the low $K_{fs}$ at 45 cm depth. However, the trend of decreasing $K_{fs}$ with depth was not



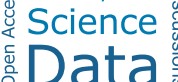

consistent across all field sites, as shown by the two arable fields with free-draining loamy soils underlain by Chalk. The field with controlled traffic exhibited a decrease in average $K_{fs}$ with depth for infield sampling locations, but for trafficked locations average $K_{fs}$ was higher at 45 cm depth compared to 25 cm depth, presumably due to surface ground pressure exerted by vehicle wheeling. This pattern was also observed in the field that was managed conventionally (without controlled traffic), however to a smaller extent, as both depths had relatively similar ranges of $K_{fs}$ (median of ~$1 \times 10^{-4}$ cm s$^{-1}$).


**Figure 6: Box plots of saturated hydraulic conductivity ($K_{fs}$) (cm s$^{-1}$) measurements taken at infield and trafficked locations at 25 cm depth, and also 45 cm depth where possible. Bold lines indicate median values. Note the log scale.**



## 4 Data availability

The data described within this paper are available via the UK Natural Environment Research Council (NERC) Environmental Information Data Centre (EIDC) at the following URL: https://doi.org/10.5285/a32f775b-34dd-4f31-aafa-f88450eb7a90 (Trill et al., 2022).

For the purposes of anonymous review only, this dataset may be accessed using:

Access link: https://data-package.ceh.ac.uk/data/a32f775b-34dd-4f31-aafa-f88450eb7a90 (Trill et al., 2022)

Username: reviewer

Password: review2023

## 5. Conclusions

The soil hydraulic and hydrological dataset described herein provides a valuable snapshot of soil properties on different typical geologies in the Thames catchment, UK, under different land uses and land management practices. This dataset captures spatial-temporal variations over a typical growing season, including samples replicated over space and at different soil depths. Notwithstanding the original survey aim to provide information on the impacts of land-based NFM measures on soil properties, these valuable spatial-temporal observations could also be used to help improve process representation and parameterization in hydrological and land surface models. The dataset highlights how trafficked arable field areas such as tramlines, in comparison to general infield areas, have a higher bulk density (and lower estimated porosity) near the soil surface and lower saturated hydraulic conductivity (both attributable to compaction). These trafficked areas, although forming a small proportion of the field area, will therefore have a disproportionate impact on the potential generation of surface runoff in response to storm events and likely provide rapid overland flow routes connecting runoff to the local watercourse network. This raises the challenge of how to represent such processes in hydrological models, particularly given the apparent disparity of scales. In addition to the soil physical properties, the dataset also contains information on soil moisture variation with depth at the sampled points in time, which may assist in the validation of remotely-sensed soil moisture products. Finally, although this was a snapshot survey over 2021, the impact of NFM measures on soil properties is only likely to fully emerge over a longer timescale and this therefore provides a valuable baseline reference against which future changes may be compared.



**Appendix A**



**Figure A1: Soil moisture retention curves showing changes in mean volumetric water content (VWC; cm³ cm⁻³) with increasing suction (pF) at infield and trafficked sampling points, and different depths.**

**Author contributions**

JC, JB and GO formulated the research concept and secured the research funding. JC and JB, supported by ET, designed the research methodology. ET, JR and PS collected the field survey data. JR and ET undertook the laboratory sample analysis. ET and JR determined secondary derived data. JB quality controlled the field and laboratory data. ET and JR compiled and curated the final dataset. This element of the LANDWISE project was led and managed by JB, ET and PR, supported by JC and GO.

JR prepared the manuscript initial draft, with contributions from ET and JB. JB, JR, ET and PR reviewed, edited and contributed to subsequent manuscript revisions.



**Competing interests**

The authors declare that they have no conflict of interest.

**Acknowledgements**

This LANDWISE research was supported by the UK Natural Environment Research Council (Grant Ref. NE/R004668/1) under its Natural Flood Management Research Programme. Additional support was kindly provided by the Natural Environment Research Council, UK Centre for Ecology & Hydrology and University of Reading to mitigate the impacts of the Covid19 pandemic on this research. The participation and assistance of the land owners and managers is also gratefully acknowledged, particularly their sharing of local knowledge and insights about soils and hydrology at individual survey sites.

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
