# Peer review of "Soil hydraulic and hydrological data from seven field sites in the Thames catchment, UK, 2021"

_Earth System Science Data, 2023_

## Referee Comment (RC1)

**General comments**

Overall, the paper is based on good background research, sufficient detail of the methods and techniques and a reasonably clear structure of the document. Additionally, the English is well written. However, there are some debatable interpretations of the data, which I am confident can be improved with some extra analysis and interpretation. Additionally, there are some illogical orderings of the methods and techniques, giving low priority to the key measurements of infiltration and conductivity; these shortcomings are easily curable. With sufficient improvements it should be relatively easy to revise the MS to a suitable publishable condition. The authors should be congratulated on assembling and reporting such a wide range of data to help improve predictions of drainage and flooding in the Thames catchment area.

**Specific comments**

1. The principal shortcoming of this study is the inadequate identification and interpretation of the role that drainable macroporosity has on infiltration and permeability; essentially the more efficient transit of fluids through a system of channels or fissures than a macropore system dominated by vughs. Vughs are more isolated volumes of pores than channels and fissures in soils of moderate to high clay content and are only relatively efficient in very sandy soils where the main mineral particles meet at small contact points and allow general rapid transfers of fluids between them, i.e. sands generally drain faster than most heavier textured soil in the same drainable landscape position.

This figure from Blackwell et al. graphically summarises these concepts of relative macropore organisation.

[Figure]

**Fig. 4.** The $O{:}\varepsilon$ characteristic for individual samples of ameliorated Lemnos loam with channels from old lucerne roots (⊠) or many coarse occluded vughs (□) and for Mundiwa clay loam B horizon ameliorated only by surface gypsum applications (+gypsum). Mean values (from Fig. 3) for Lemnos loam and Mundiwa clay loam of unameliorated soil and ameliorated soil without large channels or vughs are also shown together with intrinsic permeability ($k_a$) isolines.

Thus, as seen from Fig. 4, the occurrence of channel and fissure type macroporosity (fissures are often a feature of gypsum treated sodic soils) instead of vughs can easily lead to ten-fold or more increase in intrinsic

permeability (a definition independent of fluid viscosity that can be applied to water or gas). Further details can be found in the full paper ( European Journal of Soil Science 41(2):215 – 228)).

"A well prep brush attachment was used to lightly roughen the walls of the well hole and to remove any soil smearing that may have occurred during the augering process" (quote from line 240 of the ms), thus access to the local microporosity for the well permeameters should have been adequate at each site.

2. However, the data quality analysis reveals some strategies that could exclude some very high and very low real values, which could reflect the relative influence of extremes of macropore organisation. Possible problems are in **red text**.

*Infiltration measurements underwent Quality Control to categorise each measurement using the following data flags which are stored as "Infiltration_1_QC_Flag" and "Infiltration_2_QC_Flag" in the dataset:*

 • *"Good" = Where no apparent issues with the measurement were identified.*

• *"Invalid" = Where the measurement gave values that were not physically plausible (e.g. negative values). These values have been removed from the dataset.* **(this is sensible)**

• ***"A"* = Where the change in infiltration rate over time was observed to be notably unsteady (e.g. where plots of cumulative infiltration over time showed sudden/rapid changes).** This may have excluded some real effects of the interaction between the soil chemistry and the ionic strength of the infiltrating water.

"B" = Where < 15 ml water infiltrated during the measurement (the Mini Disk infiltrometer 280 manual states that **accurate calculation requires at least 15 ml of water to be infiltrated during each measurement). (This may exclude some real but very low values.)**

• **"C" = Where calculated *Kunsat* values were unusually high. This was determined by comparing the measured value against typical values + 3 SD (i.e. the 99.7% upper bound of the distribution) from Carsel and Parrish (1988). It is important to note that datapoints with this QC flag may in fact be correct and potentially reflect the novel soil state/structure/management at the time of measurement**. This may exclude some real and very high values and is recognised by the authors in the underlined text.

Saturated hydraulic conductivity measurements underwent QC to categorise each measurement using the following flags which are stored in the "Guelph_Permeameter_QC_Flag" column of the dataset:

• "Good" = Where no apparent issues with the measurement were identified.

• "Invalid" = Where measurements gave values that were not physically plausible, e.g. negative values or **"alpha"values outside of the valid range of 0.01-0.5 cm-1** (Soilmoisture Equipment Corp., 2012). These values have been removed from the dataset. **(This may exclude some real but very low or high values.)**

In addition, the "Guelph_Permeameter_notes" column indicates whether the double head method or the mean of two single head measurements was used for deriving *Kfs* for each measurement. The double head method is more accurate and is therefore provided in preference. However, **sometimes the data generated physically inadmissible values when using the double head method** and in this case the data was instead used in two separate single head measurement calculations. The results of the two single head measurements were then averaged. Further details are given in the Guelph Permeameter operating instructions (Soilmoisture Equipment Corp., 2012). **This needs a more detailed explanation to clarify the 'physical inadmissibility'.**

**I strongly suggest the authors review the excluded values of permeability & conductivity and seek empirical reasons for their inclusion as they may be a consequence of extreme types of macroporosity and the ionic interactions with the tap water being used in the infiltrometers. It may also be feasible to enable an experienced soil surveyor to inspect the sites to enable a macropore identification method to be used.** Such as by Graham Shepherds visual field techniques (https://www.bioagrinomics.com/visual-soil-assessment)**.**

3. There are some possible shortcomings of using a standard particle density to calculate total air-filled porosity. Line 180 assumed to be 2.65 g cm-3 , as commonly used in soil science.

For example, from the Nigerian Journal of Soil Science. SOME PHYSICAL PROPERTIES OF SOILS OVERLYING LIMESTONE PARENT MATERIAL IN SOUTHEASTERN NIGERIA, Aki, E. E. and Antigha, N.R.B. The bulk density of the topsoils ranged between 1.20 and 1.62g/cm3' for subsoils 1.33 and 1.82g/cm3 **Particle densities ranged between 2.42 and 3.10g/cm3 respectively** and total porosity for the surface and subsurface ranged between 49.7 and 50% respectively for all the soils.

I have tried to translate the data and possible errors of calculating total porosity with a standard value. Here is a summary of the analysis.

| Soil porosity was estimated using total porosity = 1- (dry bulk density/particle density) | | | | | | | |
|---|---|---|---|---|---|---|---|
| | from paper | from range in literature | only limestone | uniform p density | possible p density | | |
| | g/cm3 | g/cm3 | g/cm3 | | | | |
| example parent material | dry bulk density @ 50cm dep | uniform particle density | possible particle density | estimated total porosity cm3/cm3 | estimated total porosity cm3/cm3 | porosity error cm3/cm3 | porotity error %age |
| mudstone | 1.4 | 2.65 | 3.10 | 0.472 | 0.548 | -0.077 | -16.3 |
| | 1.4 | 2.65 | 3.10 | 0.472 | 0.548 | -0.077 | -16.3 |
| | 1.4 | 2.65 | 2.42 | 0.472 | 0.421 | 0.050 | 10.6 |
| | 1.4 | 2.65 | 2.42 | 0.472 | 0.421 | 0.050 | 10.6 |
| limestone | 1.4 | 2.65 | 3.10 | 0.472 | 0.548 | -0.077 | -16.3 |
| | 1.4 | 2.65 | 3.10 | 0.472 | 0.548 | -0.077 | -16.3 |
| | 1.4 | 2.65 | 2.42 | 0.472 | 0.421 | 0.050 | 10.6 |
| | 1.4 | 2.65 | 2.42 | 0.472 | 0.421 | 0.050 | 10.6 |
| | 1.4 | 2.65 | 2.42 | 0.472 | 0.421 | 0.050 | 10.6 |
| sand | 1.8 | 2.65 | 3.10 | 0.321 | 0.419 | -0.099 | -30.7 |
| | 1.8 | 2.65 | 3.10 | 0.321 | 0.419 | -0.099 | -30.7 |
| | 1.8 | 2.65 | 2.42 | 0.321 | 0.256 | 0.065 | 20.1 |
| | 1.8 | 2.65 | 2.42 | 0.321 | 0.256 | 0.065 | 20.1 |
| | 1.8 | 2.65 | 2.42 | 0.321 | 0.256 | 0.065 | 20.1 |
| | | | | | | max | 20.1 |
| | | | | | | min | -30.7 |

Thus, the net error variation is about 51%. Translating this to possible effects on infiltration (Fig 4 above) there are a range of possibilities with a range of 50% variation from values of about 0.12cm3/cm3 total porosity.

However, the relevance of this source of error in the data analysis is debateable since direct measurements of infiltration and permeability are used in the study which makes estimates of permeability from porosity relatively redundant. Additionally, the authors point out, below, other errors of estimating total porosity, adding further doubt to the need to calculate it at all.

"*It is important to note that the estimated porosity data are derived purely based on the assumed density of the soil mineral particles, without accounting for the proportion or density of any soil organic matter present. Therefore, estimated porosities for the soil surface samples are likely to be less accurate than the samples at greater depth due to the greater influence of organic matter in the topsoil. However, the LANDWISE Broadscale dataset provides adjusted estimates of soil surface porosities which also take organic matter content into account (Blake et al., 2022).*" I suggest the authors consider how much detail of total porosity estimation is required in the paper, unless such estimations are required in the overland flow models currently in use.

4. Benefits of Controlled Traffic Farming (CTF). The authors identify that "*The dataset highlights how trafficked arable field areas such as tramlines, in comparison to general infield areas, have a higher bulk density (and lower estimated porosity) near the soil surface and lower saturated hydraulic conductivity (both attributable to compaction). These trafficked areas, although forming a small proportion of the field area, will therefore have a disproportionate impact on the potential generation of surface runoff in response to storm events and likely provide rapid overland flow routes connecting runoff to the local watercourse network. This raises the challenge of how to represent such processes in hydrological models, particularly given the apparent disparity of scales*".

This is worth providing more emphasis in the conclusions since the introduction states "*There is evidence to suggest that applying such practices can help to restore soil structure, increase water holding capacity and macropore density, and reduce bulk density (McHugh et al., 2009)*". A common practical consequence of CTF in higher rainfall environments is soil erosion of the tramlines with sufficient downhill slope and length. (e.g. Saggau, P. Kuhwald, M. and Duttman, R. Effects of contour farming and tillage practices on soil erosion processes in a hummocky watershed. A model-based case study highlighting the role of tramline tracks. Catena, vol 228 ). Such tramline erosion has been a major concern for CTF adoption by some farmers, even leading to dis-adoption in extreme cases. Thus strong reference to these issues by the authors in this paper may help to encourage and justify further modelling of such effects to minimise the problem by developing improved future modelling.

5. The value of soil moisture retention curves. These allow *"comparisons to be made between the different agricultural management practices and their potential influence on soil hydraulic properties"*. Further analysis of the differences in the moisture retention curves comparing trafficked and untrafficked zones would be beneficial to draw out more value from the data, as well as suggestions on how this difference can be applied to overland flow models.

Technical corrections (typing errors etc)

1. The sequence of explanation of methods in the abstract and methods could be improved. In the title the information is presented as "*Soil hydraulic and hydrological data*", but in the methods section '*Measurements (n = 1300) included soil bulk density, estimated porosity, soil moisture and soil moisture retention, surface infiltration rate, and saturated hydraulic conductivity*.' hydraulic properties (Infiltration and conductivity) are explained after other methods. The order should be in reverse to match the priorities in the title. It should be relatively easy to re-order the text re methods in the Abstracts and the main materials and methods sections.

2. The original data, accessed from the link, is in an excel spreadsheet and should be understandable to most students and researchers. The overview of the data, locations and methods etc, is well explained both in adobe format and Word format. However, the spreadsheet itself could be less clumsy to use if the row of column headings and the first column were locked to make the location of the data much easier to understand.

3. Reference to footnote g in table 1 shows no explanation of HCl manual methodology.

4. land-based Natural Flood Management (NFM measures). NFM is not defined in the beginning of the Conclusions section and is advisable to be clarified for ease of reading the paper.

---

## Referee Comment (RC2)

General comments

The presented manuscript and corresponding dataset provide a useful survey of soil physical and hydraulic properties under several agricultural management practices and soil types within the Thames catchment, UK. I would like to congratulate the team involved for their sampling efforts and their contribution to environmental studies relevant to flood risk assessment and other significant hydrological processes. The manuscript is clear, the dataset is in good shape, and the metadata provides detailed descriptions. Before considering for publication, I suggest a minor revision to address some methodological aspects as well as clarifying decisions made for data quality control. Please find my comments and suggestions below.

Specific comments:

Table 3 – Why did not the authors make samples of Bd, VWC, Moisture retention, and Ksat in UN areas?

L.184 – The use of an assumed particle density of mineral soil (2.65 g/cm³) is often a good approximation, but I wonder why the authors did not consider this variable as part of their laboratory estimations. The soils exhibited different land cover, bulk density, geological formations, among other factors, which would suggest it is likely to find differences in particle density. I suggest the authors provide information on the uncertainty of adjusted estimates (L.315) and the impacts of assuming this fixed particle density.

L.190 – The dataset includes a limited number of samples of soil water retention properties covering the wet range (0-2 pF), which primarily describes larger-radius pore volumes (macropores and wide-coarse pores) related to structural porosity. Including information in this range is beneficial for the purposes of this paper due to its relevance to rainfall/runoff partitioning and drainage. Therefore, elaboration on these observations should be emphasised and not relegated to the Appendix. For instance, focus not only on the steepness of the observations (L.332) but also on the magnitudes and uncertainties encountered. To provide a more informative plot, please add error bars in Figure A1.

L.278 – "A" is used to flag unsteady infiltration rates, which the authors describe as "sudden/rapid changes" in the infiltration rate. Could these values be realistic (and thus flagged as "good") due to the interaction of physicochemical properties, such as a water repellency breakdown?

L.282 – The reasoning behind the "C" classification seems ambiguous. As the authors state, these values might indeed be correct and potentially related to "*a novel soil state/structure/management at the moment of sampling*". Hydraulic conductivity is a highly variable non-linear property, so extreme values may often be encountered. For instance, it is common to find biopores created by earthworms, resulting in unusually high values. I think this could be a realistic representation of the spatial heterogeneity of the soil pore system.

L.294 – Please include details for what is considered "physically inadmissible."

L.385. The authors interpret and derive conclusions that are not sustained by statistical analysis: "*The dataset highlights how trafficked arable field areas such as tramlines, in comparison to general infield areas, have a higher bulk density (and lower estimated porosity) near the soil surface and lower saturated hydraulic conductivity (both attributable to compaction). These trafficked areas, although forming a small proportion of the field area, will therefore have a disproportionate impact on the potential generation of surface runoff in response to storm events and likely provide rapid overland flow routes connecting runoff to the local watercourse network*". I suggest avoiding statements of this kind if the aim of this manuscript is to present and describe a dataset.

---

## Author Comment (AC1)

**Author response to referee comments on 'essd-2023-402'**

The authors thank both referees for their constructive and detailed comments and suggestions on the manuscript. We have taken these into consideration and amended the manuscript to address the issues raised and add further detail where this was needed.

We (AC) have responded to each of the individual comments made by referee 1 (RC1) and referee 2 (RC2) below. The line numbers included in parentheses refer to the clean copy of the manuscript (without tracked changes).

Referee 1 comments:

RC1: 1. The principal shortcoming of this study is the inadequate identification and interpretation of the role that drainable macroporosity has on infiltration and permeability; essentially the more efficient transit of fluids through a system of channels or fissures than a macropore system dominated by vughs. Vughs are more isolated volumes of pores than channels and fissures in soils of moderate to high clay content and are only relatively efficient in very sandy soils where the main mineral particles meet at small contact points and allow general rapid transfers of fluids between them, i.e. sands generally drain faster than most heavier textured soil in the same drainable landscape position. This figure from Blackwell et al. graphically summarises these concepts of relative macropore organisation.

[Figure]

Fig. 4. The $O{:}\varepsilon$ characteristic for individual samples of ameliorated Lemnos loam with channels from old lucerne roots (⊠) or many coarse occluded vughs (□) and for Mundiwa clay loam B horizon ameliorated only by surface gypsum applications (+ gypsum). Mean values (from Fig. 3) for Lemnos loam and Mundiwa clay loam of unameliorated soil and ameliorated soil without large channels or vughs are also shown together with intrinsic permeability ($k_a$) isolines.

Thus, as seen from Fig. 4, the occurrence of channel and fissure type macroporosity (fissures are often a feature of gypsum treated sodic soils) instead of vughs can easily lead to ten-fold or more increase in intrinsic permeability (a definition independent of fluid viscosity that can be applied to water or gas). Further details can be found in the full paper (European Journal of Soil Science 41(2):215 – 228)). "A well prep brush attachment was used to lightly roughen the walls of the well hole and to remove any soil smearing that may have occurred during the augering process" (quote from line 240 of the ms), thus access to the local microporosity for the well permeameters should have been adequate at each site.

AC: Recognition of the role of macroporosity has now been incorporated into the 'Infiltration Rate' (Line 318) and 'Saturated Hydraulic Conductivity' (Line 330) sections, including reference to Blackwell et al. (1990). A sentence has also been added to the Conclusions section to highlight this (Line 407).

RC1: 2. However, the data quality analysis reveals some strategies that could exclude some very high and very low real values, which could reflect the relative influence of extremes of macropore organisation. Possible problems are in **red text**.

*Infiltration measurements underwent Quality Control to categorise each measurement using the following data flags which are stored as "Infiltration_1_QC_Flag" and "Infiltration_2_QC_Flag" in the dataset:*

- *"Good" = Where no apparent issues with the measurement were identified.*
- *"Invalid" = Where the measurement gave values that were not physically plausible (e.g. negative values). These values have been removed from the dataset.* **(this is sensible)**
- *"A" = **Where the change in infiltration rate over time was observed to be notably unsteady (e.g. where plots of cumulative infiltration over time showed sudden/rapid changes).*** **This may have excluded some real effects of the interaction between the soil chemistry and the ionic strength of the infiltrating water.**
- *"B" = Where < 15 ml water infiltrated during the measurement (the Mini Disk infiltrometer 280 manual states that **accurate calculation requires at least 15 ml of water to be infiltrated during each measurement).** **(This may exclude some real but very low values.)**
- ***"C" = Where calculated Kunsat values were unusually high. This was determined by comparing the measured value against typical values + 3 SD (i.e. the 99.7% upper bound of the distribution) from Carsel and Parrish (1988). It is important to note that datapoints with this QC flag may in fact be correct and potentially reflect the novel soil state/structure/management at the time of measurement.*** **This may exclude some real and very high values and is recognised by the authors in the underlined text.**

*Saturated hydraulic conductivity measurements underwent QC to categorise each measurement using the following flags which are stored in the "Guelph_Permeameter_QC_Flag" column of the dataset:*

- *"Good" = Where no apparent issues with the measurement were identified.*
- *"Invalid" = Where measurements gave values that were not physically plausible, e.g. negative values or **"alpha" values outside of the valid range of 0.01-0.5 cm-1** (Soilmoisture Equipment Corp., 2012). These values have been removed from the dataset.* **(This may exclude some real but very low or high values.)**

*In addition, the "Guelph_Permeameter_notes" column indicates whether the double head method or the mean of two single head measurements was used for deriving Kfs for each measurement. The double head method is more accurate and is therefore provided in preference. However, **sometimes the data generated physically inadmissible values when using the double head method** and in this case the data was instead used in two separate single head measurement calculations. The results of the two single head measurements were then averaged. Further details are given in the Guelph Permeameter operating instructions*

*(Soilmoisture Equipment Corp., 2012)*. **This needs a more detailed explanation to clarify the 'physical inadmissibility'.**

**I strongly suggest the authors review the excluded values of permeability & conductivity and seek empirical reasons for their inclusion as they may be a consequence of extreme types of macroporosity and the ionic interactions with the tap water being used in the infiltrometers. It may also be feasible to enable an experienced soil surveyor to inspect the sites to enable a macropore identification method to be used.** Such as by Graham Shepherds visual field techniques (https://www.bioagrinomics.com/visual-soil-assessment)

AC: It is important to note that that data flagged with Quality Control letters have been retained within the published dataset. Data were only removed in the case of invalid (physically implausible) measurements, which in total removed 3 infiltration measurements and 1 saturated hydraulic conductivity measurement. However, we appreciate that the other flagged data may be real have amended the manuscript to emphasise this point to data users (Line 288). We have included suggestions of empirical reasons for the QC flagged measurements. We have also clarified what is meant by "physically inadmissible" values and why these occurred, with reference to the Guelph Permeameter manual. Unfortunately, we are unable to carry out further fieldwork for macropore identification at this stage as the LANDWISE project and associated funding ended in 2022.

RC1: 3. There are some possible shortcomings of using a standard particle density to calculate total air-filled porosity. Line 180 assumed to be 2.65 g cm-3 , as commonly used in soil science. For example, from the Nigerian Journal of Soil Science. SOME PHYSICAL PROPERTIES OF SOILS OVERLYING LIMESTONE PARENT MATERIAL IN SOUTHEASTERN NIGERIA, Aki, E. E. and Antigha, N.R.B. The bulk density of the topsoils ranged between 1.20 and 1.62g/cm3' for subsoils 1.33 and 1.82g/cm3 **Particle densities ranged between 2.42 and 3.10g/cm3 respectively** and total porosity for the surface and subsurface ranged between 49.7 and 50% respectively for all the soils.

I have tried to translate the data and possible errors of calculating total porosity with a standard value. Here is a summary of the analysis.

| Soil porosity was estimated using total porosity = 1- (dry bulk density/particle density) | | | | | | |
| --- | --- | --- | --- | --- | --- | --- |
| | from paper | from range in literature | only limestone | | | |
| | g/cm3 | g/cm3 | g/cm3 | uniform p density | possible p density | |
| example parent material | dry bulk density @ 50cm dep | uniform particle density | possible particle density | estimated total porosity cm3/cm3 | estimated total porosity cm3/cm3 | porosity error cm3/cm3 | porotity error %age |
| mudstone | 1.4 | 2.65 | 3.10 | 0.472 | 0.548 | -0.077 | -16.3 |
| | 1.4 | 2.65 | 3.10 | 0.472 | 0.548 | -0.077 | -16.3 |
| | 1.4 | 2.65 | 2.42 | 0.472 | 0.421 | 0.050 | 10.6 |
| | 1.4 | 2.65 | 2.42 | 0.472 | 0.421 | 0.050 | 10.6 |
| limestone | 1.4 | 2.65 | 3.10 | 0.472 | 0.548 | -0.077 | -16.3 |
| | 1.4 | 2.65 | 3.10 | 0.472 | 0.548 | -0.077 | -16.3 |
| | 1.4 | 2.65 | 2.42 | 0.472 | 0.421 | 0.050 | 10.6 |
| | 1.4 | 2.65 | 2.42 | 0.472 | 0.421 | 0.050 | 10.6 |
| | 1.4 | 2.65 | 2.42 | 0.472 | 0.421 | 0.050 | 10.6 |
| sand | 1.8 | 2.65 | 3.10 | 0.321 | 0.419 | -0.099 | -30.7 |
| | 1.8 | 2.65 | 3.10 | 0.321 | 0.419 | -0.099 | -30.7 |
| | 1.8 | 2.65 | 2.42 | 0.321 | 0.256 | 0.065 | 20.1 |
| | 1.8 | 2.65 | 2.42 | 0.321 | 0.256 | 0.065 | 20.1 |
| | 1.8 | 2.65 | 2.42 | 0.321 | 0.256 | 0.065 | 20.1 |
| | | | | | | max | 20.1 |
| | | | | | | min | -30.7 |

Thus, the net error variation is about 51%. Translating this to possible effects on infiltration (Fig 4 above) there are a range of possibilities with a range of 50% variation from values of about 0.12cm3/cm3 total porosity.

However, the relevance of this source of error in the data analysis is debateable since direct measurements of infiltration and permeability are used in the study which makes estimates of

permeability from porosity relatively redundant. Additionally, the authors point out, below, other errors of estimating total porosity, adding further doubt to the need to calculate it at all.

*"It is important to note that the estimated porosity data are derived purely based on the assumed density of the soil mineral particles, without accounting for the proportion or density of any soil organic matter present. Therefore, estimated porosities for the soil surface samples are likely to be less accurate than the samples at greater depth due to the greater influence of organic matter in the topsoil. However, the LANDWISE Broadscale dataset provides adjusted estimates of soil surface porosities which also take organic matter content into account (Blake et al., 2022)."* **I suggest the authors consider how much detail of total porosity estimation is required in the paper, unless such estimations are required in the overland flow models currently in use.**

AC: We believe that it is still valuable to include the porosity estimations as part of the dataset/paper, particularly as water retention at saturation data are not available for some of the samples. Additional detail on the uncertainty of derived porosity values has been added to the text (Line 362), including estimates of error based on the data from Aki and Antigha (2015). A supplementary spreadsheet (Supplement 1) showing the potential uncertainties around the derived porosity values is now included alongside the manuscript.

RC1: 4. Benefits of Controlled Traffic Farming (CTF). The authors identify that *"The dataset highlights how trafficked arable field areas such as tramlines, in comparison to general infield areas, have a higher bulk density (and lower estimated porosity) near the soil surface and lower saturated hydraulic conductivity (both attributable to compaction). These trafficked areas, although forming a small proportion of the field area, will therefore have a disproportionate impact on the potential generation of surface runoff in response to storm events and likely provide rapid overland flow routes connecting runoff to the local watercourse network. This raises the challenge of how to represent such processes in hydrological models, particularly given the apparent disparity of scales".*

This is worth providing more emphasis in the conclusions since the introduction states *"There is evidence to suggest that applying such practices can help to restore soil structure, increase water holding capacity and macropore density, and reduce bulk density (McHugh et al., 2009)".* A common practical consequence of CTF in higher rainfall environments is soil erosion of the tramlines with sufficient downhill slope and length. (e.g. Saggau, P. Kuhwald, M. and Duttman, R. Effects of contour farming and tillage practices on soil erosion processes in a hummocky watershed. A model-based case study highlighting the role of tramline tracks. Catena, vol 228 ). Such tramline erosion has been a major concern for CTF adoption by some farmers, even leading to dis-adoption in extreme cases. Thus strong reference to these issues by the authors in this paper may help to encourage and justify further modelling of such effects to minimise the problem by developing improved future modelling.

AC: The issue of soil erosion from tramlines has now been emphasised in the Conclusions (Line 415), including reference to Saggau et al., 2023.

RC1: 5. The value of soil moisture retention curves. These allow "*comparisons to be made between the different agricultural management practices and their potential influence on soil hydraulic properties*". Further analysis of the differences in the moisture retention curves comparing trafficked and untrafficked zones would be beneficial to draw out more value from the data, as well as suggestions on how this difference can be applied to overland flow models.

AC: The soil moisture retention curve figure has been revised to aid visual comparisons between the treatments, and trafficked and untrafficked zones. Accompanying text has been added to highlight some of these comparisons (Line 385). Please also see our response to Referee 2 on this point.

RC1: **Technical corrections (typing errors etc)** 1. The sequence of explanation of methods in the abstract and methods could be improved. In the title the information is presented as "*Soil hydraulic and hydrological data*", but in the methods section '*Measurements (n = 1300) included soil bulk density, estimated porosity, soil moisture and soil moisture retention, surface infiltration rate, and saturated hydraulic conductivity.*' hydraulic properties (Infiltration and conductivity) are explained after other methods. The order should be in reverse to match the priorities in the title. It should be relatively easy to re-order the text re methods in the Abstracts and the main materials and methods sections.

AC: We have re-ordered the manuscript (Abstract, Introduction, Methodology, Data sections) accordingly to reflect the priorities in the title.

RC1: 2. The original data, accessed from the link, is in an excel spreadsheet and should be understandable to most students and researchers. The overview of the data, locations and methods etc, is well explained both in adobe format and Word format. However, the spreadsheet itself could be less clumsy to use if the row of column headings and the first column were locked to make the location of the data much easier to understand.

AC: The original dataset is provided as a comma separated values (.csv) file. Whilst this opens in Microsoft Excel as default, the .csv file does not have the functionality to lock columns/rows. We have amended the Data Availability section (Line 398) to include a recommend that users lock the row of column headings and first column.

RC1: Reference to footnote g in table 1 shows no explanation of HCl manual methodology.

AC: We have elaborated on the HCl test in the footnote (Line 103) and referred readers to Blake et al. 2022 which provides details on the methodology in the Supporting Information.

RC1: 4. land-based Natural Flood Management (NFM measures). NFM is not defined in the beginning of the Conclusions section and is advisable to be clarified for ease of reading the paper.

AC: We have now also defined the NFM acronym in the Conclusions section (Line 405) to improve readability.

Referee 2 comments:

RC2: Table 3 – Why did not the authors make samples of Bd, VWC, Moisture retention, and Ksat in UN areas?

AC: Sampling design was in part driven by the data requirements of a modelling work package of the LANDWISE project. The modelling focussed on infield areas and so sampling effort was also focussed in these areas as project resources did not allow us to sample all soil properties in all locations. Furthermore, sampling focussed on infield measurements to align with transects of electrical resistivity tomography (ERT) measurements taken by project collaborators (these data are contained in a separate published dataset).

RC2: L.184 – The use of an assumed particle density of mineral soil (2.65 g/cm$^3$) is often a good approximation, but I wonder why the authors did not consider this variable as part of their laboratory estimations. The soils exhibited different land cover, bulk density, geological formations, among other factors, which would suggest it is likely to find differences in particle density. I suggest the authors provide information on the uncertainty of adjusted estimates (L.315) and the impacts of assuming this fixed particle density.

AC: Detail has been added to mention the impact of the particle density assumption, with reference to over-estimation of porosity for woodland soils with higher proportions of organic matter. Please also see our response to Referee 1 (comment 3).

RC2: L.190 – The dataset includes a limited number of samples of soil water retention properties covering the wet range (0-2 pF), which primarily describes larger-radius pore volumes (macropores and wide-coarse pores) related to structural porosity. Including information in this range is beneficial for the purposes of this paper due to its relevance to rainfall/runoff partitioning and drainage. Therefore, elaboration on these observations should be emphasised and not relegated to the Appendix. For instance, focus not only on the steepness of the observations (L.332) but also on the magnitudes and uncertainties encountered. To provide a more informative plot, please add error bars in Figure A1.

AC: Figure A1 has been moved from the Appendix into the main text describing the soil water retention properties of the samples (now Figure 7). The figure has also been modified to add error bars as suggested; this required changing the figure to boxplots and separating the data into different panels to prevent overlapping error bars. Additional detail has been added to the text (Line 384) to elaborate on the degrees of uncertainties in the data.

RC2: L.278 – "A" is used to flag unsteady infiltration rates, which the authors describe as "sudden/rapid changes" in the infiltration rate. Could these values be realistic (and thus flagged as "good") due to the interaction of physicochemical properties, such as a water repellency breakdown?

AC: Within the Data Quality Control section we have now explained that flagged datapoints may in fact be realistic values due to a number of reasons (Line 288). A sentence has been added to state that it is ultimately for the data user to decide if measurements should be included/excluded based on the information provided. The QC process followed the guidance of field instrument manuals to keep a consistent approach. Please also see our response to Referee 1 (comment 2).

RC2: L.282 – The reasoning behind the "C" classification seems ambiguous. As the authors state, these values might indeed be correct and potentially related to "*a novel soil state/structure/management at the moment of sampling*". Hydraulic conductivity is a highly variable non-linear property, so extreme values may often be encountered. For instance, it is common to find biopores created by earthworms, resulting in unusually high values. I think this could be a realistic representation of the spatial heterogeneity of the soil pore system.

AC: Additional explanation has been added to the manuscript to include reasons why these datapoints may be correct (Line 287). Flagged measurements are retained within the dataset to allow data users to include/exclude the data as they choose. Please see our response to Referee 1 (comment 2).

RC2: L.294 – Please include details for what is considered "physically inadmissible."

AC: Additional explanation has been added to detail how "physically inadmissible" values were determined (Line 303). See the author response to Referee 1 (comment 2).

RC2: L.385. The authors interpret and derive conclusions that are not sustained by statistical analysis: "*The dataset highlights how trafficked arable field areas such as tramlines, in comparison to general infield areas, have a higher bulk density (and lower estimated porosity) near the soil surface and lower saturated hydraulic conductivity (both attributable to compaction). These trafficked areas, although forming a small proportion of the field area, will therefore have a disproportionate impact on the potential generation of surface runoff in response to storm events and likely provide rapid overland flow routes connecting runoff to the local watercourse network*". I suggest avoiding statements of this kind if the aim of this manuscript is to present and describe a dataset.

AC: The wording in this concluding paragraph has been amended to show that these statements are hypotheses that require further research/testing rather than conclusions drawn directly from the dataset (Line 413). The dataset indicates that these differences exist, but further work is required to disentangle and attribute these using statistical analysis.